# Cholesterol Attenuates the Pore-Forming Capacity of CARC-Containing Amphipathic Peptides

**DOI:** 10.3390/ijms26020533

**Published:** 2025-01-10

**Authors:** Ilya P. Oleynikov, Alexander M. Firsov, Natalia V. Azarkina, Tatiana V. Vygodina

**Affiliations:** A. N. Belozersky Institute of Physico-Chemical Biology, M. V. Lomonosov Moscow State University, Leninskie Gory 1, Bld. 40, Moscow 119992, Russia; oleynikov.biophys@gmail.com (I.P.O.); amfamf@yandex.ru (A.M.F.); vygodina@belozersky.msu.ru (T.V.V.)

**Keywords:** CPPs, amphipathic peptides, cholesterol, CRAC/CARC motifs, pore, membrane, Δψ, uncoupling, liposomes, cytochrome oxidase

## Abstract

Artificial peptides P4, A1 and A4 are homologous to amphipathic α-helical fragments of the influenza virus M1 protein. P4 and A4 contain the cholesterol recognition sequence CARC, which is absent in A1. As shown previously, P4 and A4 but not A1 have cytotoxic effects on some eukaryotic and bacterial cells. This might be caused by the dysfunction of cholesterol-dependent cellular structures, inhibition of the respiratory chain, or disruption of the membrane. Here, we analyzed the latter hypothesis by studying the uncoupling effect of the peptides on asolectin membranes. The influence of A4 on Δψ pre-formed either by the valinomycin-dependent K^+^ diffusion or by the activity of membrane-built cytochrome *c* oxidase (CcO) was studied on (proteo)liposomes. Also, we investigated the effect of P4, A1 and A4 on liposomes loaded with calcein. It is found that A4 in a submicromolar range causes an immediate and complete dissipation of diffusion Δψ across the liposomal membrane. Uncoupling of the CcO-containing proteoliposomes requires an order of magnitude of higher peptide concentration, which may indicate the sorption of A4 on the enzyme. The presence of cholesterol in the membrane significantly weakens the uncoupling. Submicromolar A4 and P4 cause the release of calcein from liposomes, indicating the formation of membrane pores. The process develops in minutes and is significantly decelerated by cholesterol. Micromolar A1 induces pore formation in a cholesterol-independent manner. We conclude that the peptides P4, A4 and, in higher concentrations, A1 form pores in the asolectin membrane. The CARC-mediated interaction of A4 and P4 with cholesterol impedes the peptide oligomerization necessary for pore formation. The rapid uncoupling effect of A4 is apparently caused by an increase in the proton conductivity of the membrane without pore formation.

## 1. Introduction

In the last 30 years, the attention of researchers has been attracted by so-called cell-penetrating peptides (CPPs), named for their ability to pass through bilayer lipid membranes [1,2]. A large proportion of CPPs are relatively short (10–30 amino acids) water-soluble peptides enriched in cationic and amphipathic residues [3]. Positively charged residues provide the interaction of CCPs with anionic groups of membrane phospholipids. The secondary structure of amphipathic CPPs often includes an α-helix. The structure of several α-helical fragments connected by spacers is widespread.

A similar fragment was found in the M1 matrix protein of the influenza virus [4]. It is noteworthy that, in the sequences of three α-helices of the fragment (third, sixth and thirteenth), there is a “cholesterol-recognizing/interacting” motif (Cholesterol Recognition/Interaction Amino Acid Consensus, CRAC). Apparently, it is necessary for the formation of the cholesterol-rich envelope of viral particles from the host cell membrane. The CRAC sequence was originally determined for the peripheral-type benzodiazepine receptor, BPR [5,6] and was subsequently found in many cholesterol-interacting proteins. Currently, this motif is depicted as follows: -L/V-(X)_1–5_-Y/F-(X)_1–5_-R/K- [5,7,8,9,10,11,12,13]. Later on, the second variant of the cholesterol recognition/interaction motif was discovered, which is inverted relative to CRAC and, for this reason, was named CARC. In addition to the reverse orientation relative to the N- and C-termini of the sequence, CARC differs from CRAC in the nature of the central aromatic residue, which may be tryptophan. Thus, the CARC motif looks like this: -R/K-(X)_1–5_-Y/F/W-(X)_1–5_-L/V- [14,15,16]. Sometimes, both variants of the cholesterol-recognizing sequence coexist in one protein and even in one transmembrane α-helix, which makes possible the simultaneous binding of two cholesterol molecules to it [16].

In the studies [17,18], the artificially synthesized peptide P4 was investigated, which is homologous to the third and sixth α-helices from the M1 influenza virus protein. In the M1 protein, both helices contain CRAC, but, when constructing the P4 peptide, the sequence corresponding to helix 3 was inverted, and the central aromatic residue was replaced by W. In the sequence corresponding to helix 6, the aromatic residue was also replaced by W. As a result, two CARC motifs appeared in the P4 peptide (instead of CRAC): **RTKLWEMLV**ELGNMD**KAVKLWRKL**KR [17]. It was found that P4 and its derivative “Mut 4” (ST**KLWEMLV**ELGNMD**KAVKLWRKL**SR, CARC motifs are bolded) have a cholesterol-dependent effect on phagocytes [18]. Later, the toxic effect of these peptides on *Bacillus subtilis* and *Escherichia coli* cells was demonstrated, as opposed to the CARC-lacking derivative nScr (WVGMALENRKLKKDRLKVLKMLRWT) [19]. The authors presumably associated the observed cytotoxic effects with the dysfunction of cholesterol-dependent proteins and the structure disorder of cholesterol rafts, as well as with the disruption of membrane integrity due to the formation of non-selective pores composed of peptide oligomers. Indeed, the ability to form differently structured pores due to intermolecular contacts has been found in several groups of CPPs [20,21,22,23]. It should be noted that the cytotoxic effect on *Bacillus subtilis* cells could also be explained by an increase in the ionic permeability of the membrane, even without the formation of large pores. It is known that the dissipation of ΔμH^+^ across the bacilli membrane leads to blocking of the respiratory chain [24] and to autolysis [25].

Another possible explanation (not necessarily excluding the previous ones) for the toxic effect of P4-like peptides on phagocytes and bacterial cells may be associated with their specific influence on the respiratory chain. In our recent work [26], the effect of P4 and its derivatives on mitochondrial cytochrome *c* oxidase (CcO) was discovered and studied. The structures of mitochondrial and *Rhodobacter sphaeroides’* CcO include a regulatory site BABS (Bile Acid Binding Site), which has an affinity for a number of physiologically significant compounds of amphipathic nature [27,28,29]. It is a hydrophobic cavity in the catalytic subunit of the enzyme that opens both into the thickness of the membrane and into the internal aqueous environment near the mouth of the proton channel K that is involved in proton delivery to the oxygen reduction center at certain stages of the catalytic cycle. We have shown [26] that P4 interacts with the BABS, causing a reversible inhibition of the oxidase activity of solubilized CcO with K_i_ = 3 μM. In addition to P4, its variants A4 and A1 turned out to be effective inhibitors of the solubilized enzyme. Peptide A4 (which is the same as Mut4 in [18]) differs from P4 by substitutions of two positively charged residues at the N-terminus and near the C-terminus to serine but still contains two CARC motifs. Peptide A1 (which is the same as nScr in [19]) consists of the same residues as P4 but rearranged in such a way that CARC motifs are absent from the sequence. It turned out that peptides P4 and A4 inhibit CcO not only in solution but also as part of the mitochondrial membrane. It is interesting that, in the second case, inhibition began to develop during titration only after the addition of 10–15 μM of the peptide, although the K_i_ value remained virtually unchanged compared to that for the solubilized enzyme. This could be explained by high affinity binding of the inhibitor at low concentrations to unidentified groups within the mitochondrial membrane. It is noteworthy that, unlike P4 and A4, peptide A1 did not have an inhibitory effect on CcO in the mitochondrial membrane. In addition, after the incorporation of CcO into asolectin liposomes, the oxidase activity also acquired resistance to P4 and A4. The latter circumstance allowed us to further study the influence of the peptides on the coupling properties of the asolectin membrane using CcO as a Δψ generator.

The aim of this work was to investigate the ability of peptides P4, A4 and A1 to permeabilize asolectin liposomes depending on the presence of cholesterol in the membrane.

## 2. Results

### 2.1. Δψ on the Membrane of CcO-Containing Proteoliposomes

As has been recently shown, peptide A4 has a specific inhibitory effect on mitochondrial cytochrome oxidase both in solubilized form and as part of the mitochondrial membrane [26]. However, the incorporation of the enzyme into the asolectin membrane makes it completely resistant to the inhibition. The latter allowed us to study the uncoupling effect of peptide A4 on asolectin CcO-containing proteoliposomes.

The effectiveness of peptide A4 as an uncoupler was initially confirmed by determining the respiratory control of proteoliposomes. The rate of the oxidase reaction in the presence of reduced cytochrome *c* as an electron donor increased in the presence of 20 μM of peptide A4 by 2.4 times, which was comparable to the effect of 5 μM of protonophore CCCP (3.3 times).

We investigated the uncoupling effect in more detail using a potential-sensitive dye. In the presence of an oxidation substrate, CcO generates an electrical potential difference Δψ (minus inside) across the proteoliposomal membrane, which can be monitored by change in the optical absorption of safranin (Figure 1A). The experimental curves in the figure reflect Δψ generation until a steady state is reached. Except for the initial part of the curves (ca. 25% of the amplitude), this process is well approximated by an exponential function with a single time constant of 25–28 s (see the legend). The steady state value of Δψ can be further reduced by adding peptide A4.

As can be seen from curve 1, the addition of A4 to final concentrations of ≤6 μM does not cause any noticeable effect, whereas an increase in the peptide concentration to 11 μM entails a rapid (less than 10 s) drop of the spectral response to zero, which indicates the sharp dissipation of Δψ. In the presence of cholesterol in the proteoliposome membrane, the effect of A4 weakens (curve 2). In this case, the addition of 11 μM of peptide does not result in a complete, but rather an approximately twofold drop in the response.

The statistical significance of cholesterol-dependent differences in the A4 action is assessed in panel B. The addition of 11 μM of A4 causes a drop in the safranin response to 0.75% of the scale on the control sample versus 40% of the scale in the case of cholesterol-containing proteoliposomes (t = 2.97, *p* = 0.059235). The effects of 1 μM of A4 do not differ significantly between the control and cholesterol-containing samples.

### 2.2. Valinomycin-Dependent Δψ on the Membrane of K^+^-Loaded Liposomes

We then investigated the ability of A4 to dissipate Δψ generated in an alternative way, i.e., via valinomycin-mediated K^+^ ion leakage down the concentration gradient from the liposomes to the outside. This method of membrane energization is a long-term and common practice (see for example [30,31]). The relative magnitude of Δψ (minus inside) arising on the membrane was determined, as in the previous experiment, using safranin. The results are presented in Figure 2.

The kinetics of formation and subsequent dissipation of the diffusion potential difference under our conditions are shown in panel A. The observed response consists of an ascending part caused by the K^+^ ion release from the liposomes with the formation of Δψ and a descending part associated with the passive leakage of protons along the electric field into the liposomes. The increase in Δψ includes a fast unresolved phase (ca. 25–50% of the amplitude) and a single-exponential process (see the legend). The latter is significantly slower in the presence of cholesterol (curve 2, τ = 25 s) than in the control experiment (curve 1, τ = 9 s). The decay of Δψ in the control also corresponds to a single-exponential function with τ = 22 s. In the presence of cholesterol, this phase is visually 2.5–3 times slower but cannot be approximated correctly. It is essential that the maximum amplitude of the response in both samples is the same. Apparently, the cholesterol-containing membrane, due to its increased viscosity, is less permeable both for protons and for the potassium–valinomycin complex. This is consistent with the literature [32].

Then, after the formation of diffusion Δψ, peptide A4 was added to the liposomes (Figure 2B). The increase in Δψ had the same time characteristics as in the previous case (the τ value is about 9–10 s in the control and 20–25 s in the presence of cholesterol, see the legend). The addition of 0.5 μM of peptide A4 causes a rapid (less than 10 s) drop in Δψ to zero (curve 1). Thus, on the liposomes we used, the efficiency of A4 as a Δψ-dissipating agent is at least an order of magnitude higher than on CcO-containing proteoliposomes (compare with Figure 1A, curve 1).

On cholesterol-containing liposomes (curves 2 and 3 in Figure 2B), significantly higher concentrations of peptide A4 are required for Δψ dissipation, as compared to the control (curve 1). Adding 1 μM, 2 μM (curve 2) or 5 μM (curve 3) of A4 causes a drop in Δψ of approximately 25%, 40% and 60%, respectively. In all cases, the response develops in 5–10 s.

### 2.3. Calcein Leakage from Liposomes

The addition of submicromolar concentrations of peptide A4 to calcein-loaded liposomes results in an increase in fluorescence indicating the release of calcein into the external medium (Figure 3A).

The change in fluorescence in response to the addition of A4 includes a time-unresolved fast phase, the amplitude of which is a few percentages of the scale value (usually about 5–7%). The shape of the rest of the response probably reflects the multi-stage nature of the pore formation process. The effect depends on the concentration of the peptide. In the control experiment (curve 1), the addition of 0.2 μM of A4 causes a slowing increase in fluorescence (a response with an amplitude of 25% of the final one develops within 4.5 min after the addition). When the peptide concentration is increased to 0.5 μM, a 100% response is achieved within 80 s. Thus, the concentration of A4 required for pore formation is in the same (submicromolar) range as that required for the dissipation of the K^+^-diffusion Δψ (see curve 1 in Figure 2B), although pore formation develops more slowly than the reset of Δψ (2 min vs. <10 s). It should be noted that the kinetics observed in our experiments (Figure 3) are consistent with the literature data on the formation of pores sufficient for the release of calcein and molecules of similar size (see, for example, [33,34,35]).

In the experiment on the release of calcein from cholesterol-containing liposomes (curve 2 in Figure 3A), the rate of development and the final amplitude of responses to the addition of peptide A4 are significantly reduced compared to the control (curve 1). As shown in Panel B, these differences are statistically significant. Thus, the addition of 0.2 μM of peptide causes approximately a 10 times smaller response in the cholesterol-containing preparation than in the control (t = 4.26, *p* = 0.013048). An addition of another 0.3 μM of the peptide (final concentration is 0.5 μM) induces a response in the cholesterol-containing preparation that is about three times smaller comparing the control (t = 3.59, *p* = 0.015641).

Two amphipathic peptides related to A4 in primary sequence were also found to be effective inducers of calcein release from liposomes.

Peptide P4 differs from A4 by the presence of arginine and lysine residues in place of two flanking serines (at the N-terminus and near the C-terminus, respectively). The addition of P4 to calcein-loaded liposomes results in a fluorescent response whose rate and amplitude look to be directly dependent on the peptide concentration (Figure 3C, curves 1–3). The effects of successive additions are additive (cf. curves 2 and 3). The effective concentrations of P4 are in the submicromolar range, which are very close to those of A4 (cf. curve 1 in panel A and curve 3 in panel C). The presence of cholesterol in the liposome membrane slows down the development and reduces the amplitude of the response evoked by P4 (curve 4 in panel C, compared with control curve 3). This effect is manifested to approximately the same extent as in the case of A4 (cf. curve 2 in panel A and curve 4 in panel C). Apparently, the presence of two positive charges at the ends of the P4 peptide (instead of the uncharged residues in the A4 peptide) is insignificant either for pore formation or for interaction with cholesterol.

Peptide A1 consists of the same amino acid residues as P4 but arranged in a different order. The addition of A1 causes the release of calcein from liposomes, the effect being directly dependent on the concentration of the peptide (Figure 3D, curves 1–3). Effective concentrations of A1 are significantly higher compared to A4 and P4 and are in the micromolar range. Thus, after adding 1 μM of A1, the response reaches only 50% of the scale in 7 min, whereas, after a similar addition of peptide A4 or P4, the response reaches 100% in 1 min (cf. curve 2 in panel D with curve 1 in panel C). It is noteworthy that the presence of cholesterol in the membrane only slightly (on average by 20–40%) slows down the action of A1 (compare curves 3 and 4 in panel D). These differences are not statistically significant.

Finally, all experimental curves shown in Figure 3 were subjected to approximation. The traces of the calcein release after each addition were fitted to the function y = k(1 − exp(−x/τ)) + const or sum of such functions. The kinetic parameters obtained are given in Appendix A (see Appendix A). Although this part of the results is preliminary, some trends can be noted. The rates of responses to the A4 and P4 additions manifest a strong concentration dependence. This may indicate a positive cooperativity of the action of these peptides rooted in their oligomerization. The effect looks less pronounced in the case of A1. On cholesterol-containing liposomes, the concentration dependence of the A4 and P4 action is significantly weakened compared to the control.

## 3. Discussion

The results obtained in our work indicate that peptide A4 can effectively uncouple oxidative phosphorylation. This follows from its ability to quickly reduce the transmembrane electrical potential difference to almost zero. An alternative, though unlikely, explanation for this phenomenon could be complete membrane disruption by the amphipathic peptide, similar to that caused by detergent. We excluded such a possibility by evaluating the particle size distribution in the liposome suspension under different conditions (see Appendix A in Appendix A). In the control experiment, 99% of the volume occupied by liposomes is represented by particles with a diameter of 60–70 nm and about 200 nm (panel A). Incubation of liposomes in the presence of 10 μM of A4 causes a slight shift towards smaller sizes, with a distribution maximum in the region of 100 nm and an additional peak at 30–40 nm (panel B). This may be due to the effect of the peptide on the hydrophobic and/or charge interactions of the vesicles with each other, but there are no signs of their destruction. As a positive control, detergent was added to the suspension (panel C). This caused an immediate clearing of the suspension and a drop in the size of the remaining particles to 10 nm and below. Thus, it can be stated that peptide A4 does not cause destruction of the asolectin membrane.

In the simplest experimental system, Δψ was created on the membrane of asolectin liposomes by the diffusion of potassium ions along a concentration gradient, without any proteins embedded in the membrane. In this case, 0.5 μM of A4 in a few seconds completely reset Δψ assessed by the voltage-dependent response of safranin (see Figure 2B, curve 1). Consistent with this result, peptide A4 at submicromolar concentrations induces the release of calcein from liposomes (Figure 3A). In the second case, one can assume pore formation, with the pore structure formed by A4 molecules as oligomers. This pore formation mechanism is typical among amphipathic penetrating peptides with an α-helix secondary structure [20,21]. Previously, the α-helical conformation of the P4, A4 and A1 peptides both in the asolectin bilayer and in detergent micelles was confirmed by Circular Dichroism and by molecular dynamics [26].

It is characteristic that the same addition of peptide causes a much slower response in the calcein release experiment compared to the Δψ reset (ca. 2 min versus <10 s for 0.5 μM A4). It can be assumed that A4 increases the proton permeability of the membrane well before pores have time to form in it. Accordingly, Δψ dissipation immediately after the addition of the peptide may be associated with the passive entry of protons into liposomes along the electric field. The pore formation process requires more time and, judging by the shape of the kinetic curves, is multi-stage (Figure 3A). The ability to induce membrane ion conductivity by different mechanisms (with and without pore formation) has been demonstrated for some gramicidin A derivatives [36,37]. In our case, the formation of pores is induced, besides A4, by P4 (Figure 3C) and A1 (Figure 3D), which both are relative to A4 in the primary sequence. Notably, all three peptides contain tryptophan and also have positively charged residues near the N-terminus, which are two necessary prerequisites for pore formation in some gramicidin A derivatives [38].

The most interesting phenomenon revealed in our work seems the ability of the membrane cholesterol to modulate the pore-forming action of amphipathic peptides. An alternative explanation for our results could be a negative effect of cholesterol on the binding of peptides to the asolectin membrane, which is the first step of interaction preceding pore formation. However, we have ruled out the possibility of such a scenario (see Appendix A in Appendix A). For this purpose, asolectin liposomes either containing or not containing cholesterol were mixed with 10 μM of A4 and spanned down by centrifugation. The amount of peptide remaining in the supernatant could be estimated from the optical absorption spectrum in the 200–300 nm region. The peptide content in the supernatant decreased with increasing amount of liposomes in the sample, indicating peptide sorption by the membrane (cf. spectra 4 vs. 6 and 5 vs. 7 in Appendix A). The presence of cholesterol in the liposome membrane slightly increased the efficiency of sorption, which was manifested in a decrease in the peptide content in the supernatant by 5–10% (cf. spectra 4 vs. 5 and 6 vs. 7). Thus, the weakening of pore formation in the presence of cholesterol described in our work is obviously associated with interference in the pore formation mechanism.

Initially, cholesterol-dependent effects of P4 and A4 were described in phagocytes [17,18]. A decrease in the cholesterol content in the phagocyte membrane led to an increase in the sensitivity of the latter to the toxic effects of the indicated peptides. Although the nature of the toxicity was not established, the authors considered two main possibilities. First, CARC-containing peptides can deactivate cholesterol- and raft-dependent membrane proteins due to the binding (sequestration) of membrane cholesterol by CARC motifs. It is obvious that cholesterol-recognizing regions of peptides and cellular proteins must compete with each other for binding of their ligand. Secondly, non-selective pores may form in the membrane due to peptide oligomerization. Subsequently, similar mechanisms were proposed to explain the antimicrobial effects of P4 (dysfunction of sterol-dependent proteins) and A1 (membrane permeabilization) on *Escherichia coli* and *Bacillus subtilis* cells, respectively [19]. In our study on artificial membrane objects, asolectin liposomes, we can hardly talk about rafts or even raft-like structures. Nonetheless, we can state that membrane cholesterol modulates the membrane-related action of the CARC-containing P4 and A4 peptides (Figure 3A–C), without affecting that caused by the CARC-lacking A1 peptide (Figure 3D). Apparently, the CARC motif-mediated interaction of the peptides with cholesterol hinders their intermolecular contacts during pore formation.

To our knowledge, such an effect of cholesterol on the oligomerization of CRAC/CARC-containing penetrating peptides has not been reported so far. Among the works carried out on liposomes and devoted to the effect of cholesterol on pore formation, there are indications of a decrease in the ability to oligomerize in the presence of cholesterol for the GALA peptide [39], as well as a cholesterol-dependent decrease in the efficiency of pore formation by the polyene antibiotic nystatin [40]; however, both of these agents do not contain CRAC/CARC motifs. In contrast, for the CRAC-containing adenylate cyclase toxin, a stimulating effect of cholesterol on oligomerization and pore formation was shown [41]. In the case of the apoptogenic protein BAX, which carries the CRAC sequence in one of its α-helices, the presence of cholesterol in the liposomal membrane is not essential for pore formation [35].

We found that, in addition to slowing down pore formation, the presence of cholesterol in the membrane also weakens the uncoupling effect of peptide A4, which manifests itself as a rapid reset of Δψ (Figure 1 and Figure 2). In this case, however, the most likely explanation is associated with a decrease in the passive proton conductivity of the cholesterol-containing membrane and does not require the involvement of cholesterol–peptide interactions. Indeed, the uncoupling responses evoked by the peptide are not slowed down in the presence of cholesterol, as occurs in the calcein release experiment (Figure 3A), but only decreased in amplitude. Both experiments take place in a steady state, where the observed level of the safranin spectral response is determined by the ratio of two rates: Δψ generation (due to K^+^ diffusion or due to cytochrome oxidase activity) and proton leakage (both peptide-mediated and passive). A decrease in passive leakage will lead to an increase in the steady-state level of Δψ, which is what we observe (see curves 2 in Figure 1A and Figure 2B).

An unexpected result was a strong discrepancy between the effective concentrations of A4 as an uncoupler in two experimental systems: on liposomes with Δψ created by a valinomycin-dependent K^+^ diffusion and on proteoliposomes with incorporated cytochrome oxidase generating Δψ during the oxidase reaction. In the first case, 0.5 μM of A4 was sufficient to reset Δψ, whereas, in the second case, an order of magnitude of higher concentration of the peptide (7–11 μM) was required (see Figure 1A, curve 1). In the second case, we assume the sorption of A4 molecules on the surface of the embedded protein (cytochrome oxidase), which prevents the uncoupling effect of A4 until all parasitic binding points are saturated. In our previous experiments on mitochondria and submitochondrial particles, we obtained a very similar value of putative parasitic binding (about 10 μM) when studying the inhibitory effect of the peptide P4 on the oxidase activity (see Figure 4 in [26]). The presumably high affinity of amphipathic peptides for the membrane-bound CcO surface appears to be an interesting phenomenon that requires further study.

## 4. Materials and Methods

*Chemicals*: Safranin O, nigericin, valinomycin, carbonyl cyanide *m*-chlorophenyl hydrazone (CCCP), cholesterol, choline chloride, potassium cyanide, sodium dithionite, cholic acid, deoxycholic acid, cytochrome *c* (type III, from the equine heart), N,N,N′,N′-tetramethyl-p-phenylenediamine (TMPD), Triton X-100 and L-ascorbic acid were from Sigma-Aldrich (Saint Louis, MO, USA). β-D-Dodecyl maltoside (DM) of Sol-Grade quality was from Anatrace (Maumee, OH, USA). pH-buffers and ethylenediaminetetraacetic acid (EDTA) were from Amresco (Radnor, PA, USA). Calcein was from MP Biomedicals (Solon, OH, USA).

*Peptides*: Synthetic peptides P4, A1 and A4 (purity ≥ 99%) were purchased from Syneuro Ltd. (Moscow, Russia). The peptides were acetylated at the N-terminus and amidated at the C-terminus. Their primary sequences are as follows: A1 (Ac-WVGMALENRKLKKDRLKVLKMLRWT-NH2); A4 (Ac-STKLWEMLVELGNMDKAVKLWRKLSR-NH2); andP4 (Ac-RTKLWEMLVELGNMDKAVKLWRKLKR-NH2). For stock solutions, peptides were dissolved in DMSO (Sigma-Aldrich, Saint Louis, MO, USA)) to a final concentration of 2–10 mM and stored at +4 °C for one week.


*Cytochrome c oxidase* was purified from the mitochondria of bovine hearts by a step-wise ammonium precipitation in detergent solution (a modified method of Fowler et al. [42] described previously [43]). Hearts were purchased from the abattoir of Pushkinsky Myasnoy Dvor Ltd. (Pushkino, Moscow region, Russia) and stored on ice for 2–3 h after slaughter until the isolation procedure began. The concentration was determined from the difference absorption spectra (dithionite reduced vs. air oxidized) using molar extinction coefficient ∆ε_605–630 nm_ = 27 mM^−1^ cm^−1^. Before reconstitution into liposomes, CcO was additionally purified on a sucrose gradient as described in [44].

*Oxidase activity* was measured amperometrically with a covered Clark-type electrode as the rate of oxygen consumption during oxidation of the substrate (5 mM of ascorbate, 0.1 mM of TMPD and 10 μM of cytochrome *c*) using the Oxytherm device (Hansatech, Norfolk, UK), in a thermostatted cell at 25 °C with permanent stirring. The assays were mostly performed in a medium containing 50 mM of a Hepes/Tris buffer, pH 7.6, 0.1 mM of EDTA, 50 mM of KCl and 0.05% dodecyl maltoside.

*Reconstitution of CcO into liposomes* was performed mostly as described earlier [45]. A mixture of phospholipids from soybeans (asolectin type IIS, Sigma-Aldrich, Saint Louis, MO, USA) was used. When necessary, 20% (w/w) of cholesterol was added. The lipid/CcO ratio was 200/1, which ensured that each liposome contained no more than one enzyme molecule (the proton leakage minimization condition). A possible admixture of empty liposomes did not interfere with the subsequent studies on Δψ generation. The procedure included the step-wise removal of cholate from the sonicated lipid mixture by Bio-Beads balls (Bio-Red Laboratories, Hercules, CA, USA) followed by the overnight dialysis. Sonication was carried out using disintegrator Branson Sonifier 150 (Brookfield, CT, USA). As shown in [45], the orientation of CcO in the resulting proteoliposomes is mainly (ca. 70%) of a mitochondrial-type, with the cytochrome *c* binding side exposed outside. K^+^- loaded liposomes were prepared in a 50 mM potassium phosphate buffer, pH 7.5 without detergent, as described in [26].

*The preparation of calcein-loaded liposomes* was essentially similar, but the medium (20 mM MOPS/Tris pH 7.5, 50 mM of KCl, 0.2 mM of EDTA) was supplied with 50 mM of calcein. Before the experiment, external calcein was removed by passing the sample through a column with Sephadex G-50 coarse (Pharmacia, Uppsala, Sweden). The fluorescence of calcein was monitored at 520 nm (excitation at 490 nm) with a Panorama Fluorat 02 spectrophotometer (Lumex, Sankt Peterburg, Russia).

*The value of Δψ across the liposome membrane in the presence of valinomycin* was measured by the change in the optical absorption of safranin O (555 nm versus 523 nm) [46] using a spectrophotometer SLM Aminco DW-2000 (SLM Instruments, Urbana, IL, USA) in a dual-wavelength mode. Liposomes containing 50 mM of K-phosphate, pH 7.5 were placed in a 100-fold volume of 50 mM of HEPES/Tris pH 7.5 supplied with 80 mM of choline chloride. The formation of Δψ was triggered by the addition of valinomycin.

*The value of Δψ on the membrane of CcO-containing proteoliposomes* was also determined using safranin (see above). The presence of nigericin in the medium ensured the complete conversion of ΔμH^+^ formed across the proteoliposome membrane due to the oxidase reaction into Δψ. The direct electron donor for CcO was cytochrome *c*. Ascorbate was used to maintain cytochrome *c* in a reduced state. Measurements were carried out in a medium containing 50 mM of potassium phosphate pH 7.5, 0.2 M of sucrose, and 0.2 mM of EDTA.

*Data processing* was performed mostly using Origin Microcal software v. 7.0 and 9.0 (https://www.originlab.com, accessed on 1 September 2018). To determine the statistical significance of the difference between two groups, Student’s *t*-test was used. Mean values were compared, taking into account the mean error of the arithmetic mean and the number of measurements in groups.

## Figures and Tables

**Figure 1 ijms-26-00533-f001:**
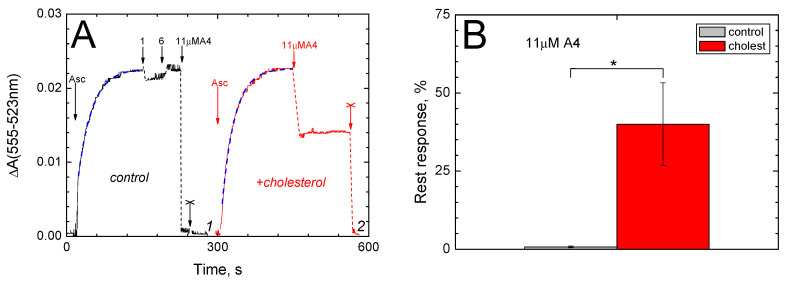
The effect of peptide A4 on Δψ generated across the membrane of CcO-containing proteoliposomes depends on the presence of cholesterol in the membrane: (**A**) The kinetics of Δψ generation. The absorption of safranin at 555 nm versus 523 nm is recorded. The proteoliposome membrane did not contain (control, black curve 1) or contained (red curve 2) cholesterol. The additions of ascorbate (Asc) to 5 mM, A4 peptide to 1 μM, 6 μM and 11 μM and cyanide to 4 mM (cross) are indicated by arrows. The experimental medium is supplied with 0.2 μM of cytochrome *c*, 0.5 μM of nigericin and 5 μM of safranin. Proteoliposomes were added up to 10 nM of CcO. Blue dash corresponds to the function: y = k(1 − exp(−x/τ)) + const, with τ = 27.8 ± 0.2 (curve 1) and τ = 24.9 ± 0.2 (curve 2). (**B**) Dissipation of Δψ by A4 depends on the presence of cholesterol in the membrane. The ordinate axis is the residual response amplitude after adding the peptide. Control (proteoliposomes without cholesterol) is shown in gray. Data for cholesterol-containing proteoliposomes are shown in red. Mean values with standard deviations are presented, as well as statistical significance of differences in mean values (*).

**Figure 2 ijms-26-00533-f002:**
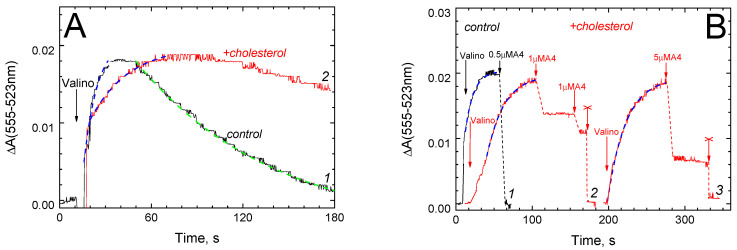
Effect of A4 on the diffusion Δψ formed across the membrane of liposomes loaded with K^+^ ions in the presence of valinomycin. Changes in Δψ are registered by the optical absorption of safranin, as in Figure 1. The liposome membrane did not contain (control, black curves 1) or contained cholesterol (red curves 2, 3). (**A**) Formation and decay of Δψ on the liposome membrane in the absence of additives. Generation of Δψ is triggered by 1 μM of valinomycin (Valino, shown by arrow). The time-resolved part of this process is approximated by the function y = k(1 − exp(−x/τ)) + const (blue dash), with τ = 9.0 ± 0.8 (curve 1) and τ = 25.0 ± 1.6 (curve 2). The decline of curve 1 is approximated by the function y = k(exp(−x/τ)) + const with τ = 21.6 ± 0.4 (green dash). (**B**) The effect of peptide A4 on the value of diffusion Δψ across the liposome membrane depends on the presence of cholesterol in the membrane. Δψ is created as in panel A. The sequential additions of 1 μM of valinomycin, peptide A4 (to final concentration of 0.5, 1 and 2 μM) and 5 μM of CCCP (cross) are indicated by arrows. The Δψ increase is approximated by the function y = k(1 − exp(−x/τ)) + const (blue dash), with the τ values of 10.9 ± 0.2 (curve 1), 22.5 ± 0.2 (curve 2) and 23.9 ± 0.3 (curve 3).

**Figure 3 ijms-26-00533-f003:**
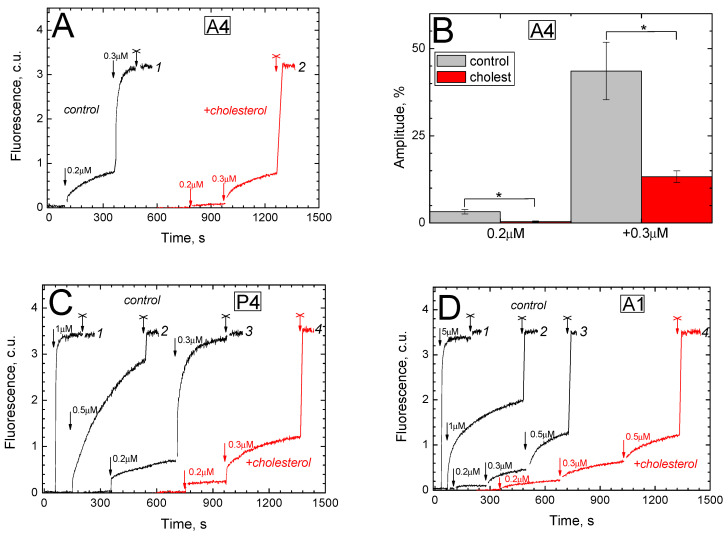
Amphipathic peptides as pore inducers in calcein-loaded liposomes. The response was determined by the increase in the fluorescence of calcein in the external environment, which indicated the leakage of calcein from the liposomes. To determine the amplitude of the complete response, the membrane is destroyed with the detergent Triton X-100 (0.1%; marked with a cross). (**A**) The effect of A4 on liposomes either not containing (control, black curve 1) or containing (red curve 2) membrane cholesterol. Successive additions of A4 are shown by arrows. (**B**) The effect of cholesterol on the release of calcein from liposomes upon the addition of A4 to final concentrations of 0.2 and 0.5 μM. Gray color—results of control experiments on cholesterol-free liposomes. Red color—results of experiments on cholesterol-containing liposomes. Mean values with standard deviations are shown, as well as the statistical significance of differences in mean (*). (**C**,**D**) The effect of peptides P4 (**C**) and A1 (**D**) on the release of calcein from liposomes either lacking (control, black curves 1–3) or containing (red curve 4) membrane cholesterol. The addition of peptides during the experiment is shown by arrows.

## Data Availability

Data are contained within this article.

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
