# Peer review of "Cholesterol Attenuates the Pore-Forming Capacity of CARC-Containing Amphipathic Peptides"

_ijms, 2025, doi:10.3390/ijms26020533_

Round 1

Reviewer 1 Report

Comments and Suggestions for Authors

Short membrane-penetrating peptides are in the center of attention of many research groups due to its relevance to viral cycle progression and putative therapeutic strategies. In their manuscript entitled “Cholesterol Attenuates the Pore-Forming Capacity of CRAC-Containing Amphipathic Peptides” Oleynikov and collaborators attempt to add a few potentially interesting datasets to supplement the current picture of the molecular mechanisms governing biological activity of such peptides and in particular the role of cholesterol-binding motifs. Generally, the manuscript is logically organized, and the obtained data are presented in appropriate context. However, while going into details some major issues (listed below) appear, which should be carefully addressed by the authors prior to publication.

-              First of all, the presented data do not directly prove the mechanism of pore formation. Could the authors prove that the observed calcein release is not the result of membrane disintegration and vesicle rupture? This could be monitored with e.g. dynamic light scattering.

-              The kinetics curves should be fitted to mathematical models – this will enable to extract all the parameters which help to directly compare the studied systems, make more consistent statements for describing results and provide more details on molecular mechanisms (see above). For sake of transparency, rates and amplitudes should be tabularized.

-              Could it be that the observed differences in the activity of a particular peptide type is linked to variations in binding capacities of various liposomal systems (with or without cholesterol)? Simple binding tests (e.g. flotation in sucrose gradient, see e.g. DOI: 10.1074/jbc.M114.602672) could help to solve this important issue.

-              It is a bit surprising that the authors focused only on A4 while measuring membrane potential. Would the other two (A1 and P4) be inefficient in this respect? 

-              The P4, A4 and A1 peptides should be introduced and their biological significance should be described in the first section. 

-              What was the yield of CoO reconstitution into vesicles (e.g. lipid/CoO ratio)?

-              Were the synthesized peptides tested for solubility/aggregation (e.g. via centrifugation and dynamic light scattering) and secondary structure (e.g. via circular dichroism)?

-              Some minor issues include: “asolectin” (instead of “azolectin”) should be used; it is commonly accepted that aa sequences are given from N- to C-terminus thus there is no need to state it in the text; statistical methods should be described; for sake of consistency it is better to substitute “ibid.” with appropriate reference;  when discussed the protonophore effect on “..the same preparation.” a reference to the data should be provided; on Fig. 2B the second portion of A4 should give 2uM concentration (not 1uM); formatting of reference list should be unified

Author Response

We are grateful to the Reviewer for carefully reading the manuscript and useful comments. Our responses are as follows.

  1. First of all, the presented data do not directly prove the mechanism of pore formation. Could the authors prove that the observed calcein release is not the result of membrane disintegration and vesicle rupture? This could be monitored with e.g. dynamic light scattering.

Answer.

- We tested the Reviewer's proposed alternative explanation. To this end, we used dynamic light scattering and evaluated the particle size distribution in the liposome suspension under different conditions. The addition of peptide A4 has only a minor effect on liposome size, which is not comparable to the catastrophic change caused by the detergent. The description of the experiment and its results are placed in section Supplementary Material (see Figure S1). In addition, in the revised version of the manuscript, the liposome disintegration hypothesis and its testing are considered in the Discussion section (highlighted in color).

  1. The kinetics curves should be fitted to mathematical models – this will enable to extract all the parameters which help to directly compare the studied systems, make more consistent statements for describing results and provide more details on molecular mechanisms (see above). For sake of transparency, rates and amplitudes should be tabularized.

Answer.

- We have fulfilled this wish of the Reviewer. In the corrected version of the manuscript, the kinetics in Figures 1 and 2 are approximated by single-exponential processes, the parameters of which are given in the Legends and in the Main text (highlighted in color). The most interesting kinetics are those associated with pore formation (Figure 3). They required a more complex approximation by the sum of 1–3 processes. The parameters (time constants and amplitudes) are given in Table S1 (see Supplementary Material). These results are mentioned at the end of the Results section (highlighted in color). We consider them to be preliminary. The detailed mechanism of pore formation by peptides A4, P4 and A1 is beyond the scope of our current communiication. It requires further research and may become the topic of our future work.

  1. Could it be that the observed differences in the activity of a particular peptide type is linked to variations in binding capacities of various liposomal systems (with or without cholesterol)? Simple binding tests (e.g. flotation in sucrose gradient, see e.g. DOI: 10.1074/jbc.M114.602672) could help to solve this important issue.

Answer.

- Thanks for this suggestion, it could be the case. To test this hypothesis, we conducted another additional experiment (see Figure S2 in Supplementary Material). Asolectin liposomes either containing or not containing cholesterol were mixed with A4 and spanned down by centrifugation. The amount of peptide remaining in the supernatant was estimated from the optical extinction. We have found that the presence of cholesterol in the liposome membrane slightly increased the efficiency of the peptide sorption to it, which was manifested in a decrease in the peptide content in the supernatant by 5-10% (cf. spactra 4 vs 5 and 6 vs 7 in Figure S2). Thus, the weakening of pore formation in the presence of cholesterol described in our work is obviously associated with interference in the pore formation mechanism rather than in the binding of the peptide to the membrane. The corresponding explanation is added into the Discussion section (highlighted in color).  

  1. It is a bit surprising that the authors focused only on A4 while measuring membrane potential. Would the other two (A1 and P4) be inefficient in this respect?

Answer.

- We have preliminary results indicating that peptides A1 and P4 effectively discharge the diffusion Dy. However, the lack of these peptides and numerous artifacts in the measurements did not allow us to draw statistically reliable conclusions about their action and the effect of cholesterol in this case. Since convincing data were obtained for peptide A4 in all types of experiments, we chose it as the main object.

  1. The P4, A4 and A1 peptides should be introduced and their biological significance should be described in the first section

Answer.

- Done.

  1. What was the yield of CoO reconstitution into vesicles (e.g. lipid/CoO ratio)?

Answer.

- The lipid/CcO ratio was 200/1 which ensured that each liposome contained no more than one enzyme molecule (the proton leakage minimization condition). Possible admixture of empty liposomes did not interfere with the subsequent studies on Dy generation. The corresponding clarification has been made in the Materials and Methods section (highlighted in color).

  1. Were the synthesized peptides tested for solubility/aggregation (e.g. via centrifugation and dynamic light scattering) and secondary structure (e.g. via circular dichroism)?

Answer.

- Previously, we studied the solubility of peptides in water, detergent solution and liposomal membrane, as well as their secondary structure (which is mainly α-helical) using circular dichroism and molecular modeling (Oleynikov et al., IJMS 2023, 24(4), 4119).

  1. Some minor issues include: “asolectin” (instead of “azolectin”) should be used; it is commonly accepted that aa sequences are given from N- to C-terminus thus there is no need to state it in the text; statistical methods should be described; for sake of consistency it is better to substitute “ibid.” with appropriate reference;  when discussed the protonophore effect on “..the same preparation.” a reference to the data should be provided; on Fig. 2B the second portion of A4 should give 2uM concentration (not 1uM); formatting of reference list should be unified.

Answer.

- Corrected.

Reviewer 2 Report

Comments and Suggestions for Authors

This is a very interesting mechanistic investigation on the impact of cholesterol-binding domain on the formation of cytotoxic pores by amphipathic peptides.

However, although the authors are aware of both CRAC and CARC consensus motifs, it is totally incomprehensible to read “due to the structural and functional similarity of the two sequences, we will henceforth use the name CRAC motif regardless of its orientation relative to the N- to C-termini of the peptide.”

Unfortunately, the studied motifs are not at all CRAC but CARC motifs and this makes a real difference that should be clearly indicated in the title and throughout the manuscript. Orientation in the membrane matters. Most importantly, the aromatic residue in the CRAC motif is exclusively Y, whereas it can be F, Y or W in the CARC motif. Thus, I strongly suggest that the authors provide an alignment of the peptide sequences and highlight the CARC motifs.

А1   WVGMALENRKLKKDRLKVLKMLRWT 

А4   STKLWEMLVELGNMDKAVKLWRKLSR 

Р4   RTKLWEMLVELGNMDKAVKLWRKLKR

With these changes, I will recommend publication.

Author Response

We are grateful to the Reviewer for carefully reading the manuscript and useful comments. Our response is as follows.

Unfortunately, the studied motifs are not at all CRAC but CARC motifs and this makes a real difference that should be clearly indicated in the title and throughout the manuscript. Orientation in the membrane matters. Most importantly, the aromatic residue in the CRAC motif is exclusively Y, whereas it can be F, Y or W in the CARC motif. Thus, I strongly suggest that the authors provide an alignment of the peptide sequences and highlight the CARC motifs.

Answer.

- The reviewer convinced us and we made appropriate changes throughout the manuscript.

Round 2

Reviewer 1 Report

Comments and Suggestions for Authors

I would like to thank the authors for addressing all the issues raised by the Reviewer. The manuscript improved a lot and now is ready for publication.

Reviewer 2 Report

Comments and Suggestions for Authors

The manuscript has been revised according to my suggestions. I recommend publication.